# The Effects of Humic Acids on the Early Developmental Stages of African Cichlids during Artificial Breeding

**DOI:** 10.3390/life13051071

**Published:** 2023-04-23

**Authors:** Silvia Ondrašovičová, František Zigo, Július Gogoľa, Zuzana Lacková, Zuzana Farkašová, Juliana Arvaiová, Viera Almášiová, Ibrahim F. Rehan

**Affiliations:** 1Department of Biology and Physiology, University of Veterinary Medicine and Pharmacy, Komenského 73, 041 81 Košice, Slovakia; 2Department of Nutrition and Animal Breeding, University of Veterinary Medicine and Pharmacy, Komenského 73, 041 81 Košice, Slovakia; 3Private Veterinary Clinic, Zvolenská Slatina SNP 367/25, 962 01 Zvolen, Slovakia; 4Department of Morphological Disciplines, University of Veterinary Medicine and Pharmacy in Košice, Komenského 73, 041 81 Košice, Slovakia; 5Department of Husbandry and Development of Animal Health, Faculty of Veterinary Medicine, Menoufia University, Shebin Alkom 32511, Egypt; 6Department of Pathobiochemistry, Faculty of Pharmacy, Meijo University, Yagotoyama 150, Tempaku-ku, Nagoya-shi 468-8503, Japan

**Keywords:** aquarium, African cichlids, artificial breeding, fry, growth, humic acids

## Abstract

The aim of this study was to compare the effect of humic acid (HA) obtained by extraction from alginate on the incubation of roes and fry development in African cichlids, *Labidochormis caeruleus*, as well as their influence on the stabilization of the physicochemical parameters of water in an aquarium during artificial breeding. The roes were obtained by extruding from a female buccal cavity immediately after fertilization. For the experiment, 4 groups of 40 roes were formed in an incubator with an artificial hatchery. Groups 1–3 were exposed to 1%, 5%, and 10% concentrations of HA, respectively. The control group C was not exposed to HA. In all groups, the mortality and size differences of the fry, as well as the temperature, pH, hardness, nitrite, and nitrate levels in the tanks, were determined during a 30-day monitoring period until the resorption of the yolk sac. The results of this study indicated the ability of HA in 5% and 10% concentrations to reduce nitrite and nitrate levels in the aquatic environment, which significantly reduced the mortality of roes and the survivability of the fry. The determination of the morphological measurements of the fry revealed an increased body length in the groups exposed to 5% and 10% HA concentrations compared to the control group by the end of the monitored period. It was also noted that the yolk sac was resorbed two days earlier in the same groups than in the control. Thus, the results showed that HAs are suitable for use in the artificial aquarium incubation of roes and fry development, which are increasingly exposed to adverse environmental factors. The knowledge obtained in this study and its transfer into practice can allow even less experienced aquarists to successfully breed aquarium fish species that could not normally be bred under artificial conditions without the addition of HA.

## 1. Introduction

Aquaculture is currently one of the most popular hobbies. The number of aquarists is increasing, and the information about fish breeding and growing aquatic plants is becoming more comprehensive. What was difficult for an experienced aquarist a few years ago, a beginner can do much faster and at a higher level today; however, some areas are still problematic due to the lack of knowledge of therapy for certain diseases as well as the incubation and rearing of the roes and fry of some types of aquarium fish. The main problem is the presence of harmful substances in aquarium water, which cause acute poisoning; a similarly frequent cause of death is the fungus on roes and fry caused by the physical and chemical changes in water that create suitable conditions for the multiplication of pathogenic agents [1,2].

One of the ways to prevent such undesirable conditions and losses during fish breeding is water treatment by means of various fossil minerals such as smectite, bentonite, lignite, leonardite, and alginite. Many of them are special substances able to detoxify the microenvironment, but, primarily, some of them contain large amounts of humates, which are important to both water fauna and flora [3,4].

Primarily, alginite appears to be a prospective clay mineral for widespread use due to its high content of humic substances. Alginite is a non-ore raw material arising from the fossilization of accumulated organic (algae) and inorganic material, especially clay (montmorillonite, illite, smectite), carbonates (dolomite, calcite, aragonite), quartz, and the amorphous modification of silicic acid in the aquatic environment. The organogenic sediment associated with oil shale arose from the action of a group of primitive yellow-green algae (*Botryococcus braunii*). The humic substances contained in alginite form a mixture of various components insoluble in water: humic acids soluble in alkalis; fulvonic acids soluble in acidic environments; humatometalic acids soluble in alcohol and acetyl bromide. Fulvonic acids (brown humic acids) and humics (black humic acids) are distinguished by the structures of their molecules and their binding to soil [5].

The use of humic acids (HA) in aquariums is still at an early stage, although experienced aquarists use them even without knowing how they affect not only the physical and chemical properties of water but also the health and immunity of fish. For this reason, several years ago, research concerning the positive or negative impacts of humic compounds and their subsequent use in aquariums for the prevention of adverse effects and the potential therapy of various diseases was initiated [6,7,8].

The results of relevant investigations indicated a whole range of beneficial properties of humic substances, such as an increase in the non-specific immune response, increased viability, and even the survival of fish under conditions that would not have allowed them to survive without the addition of humic substances. However, the available positive research involved only adult fish and no roes or fry [9,10,11,12]. They are of little relevance to advanced aquarists, who, in addition to breeding fish for pleasure, also try to reproduce various types of aquarium fish with different demands. For breeding and reproduction, they are trying to develop various artificial methods involving complex mechanisms and various factors that act on the aquarium environment [13] and are supported by the dosage of various natural substances, including HA [6].

The aim of the study was to compare the effect of HA at different concentrations on mortality and quantitative parameters of growth (body length, yolk sac resorption rate) during incubation of roes and development of fry in a selected species of aquarium fish from the Cichlidae family (*Labidochromis caerelus*) and to confirm their effects on the stabilization of physicochemical properties of water (temperature, pH, hardness, nitrite, and nitrate levels) essential for the survival of fish and successful artificial breeding.

## 2. Materials and Methods

### 2.1. Cichlids Fish, Environmental Conditions, and Reproduction

The cichlids are among the least demanding species of fish, with adult dimensions of 8–13 cm. Some species, such as *Labidochormis caeruleus*, belong to the mouthbrooding species of fish and take good care of their fry. Under natural conditions, once the female lays her roes on a flat rock, she will immediately pick them up and scoop them inside her mouth. Male cichlids of this genus have roe spots on the anal fin. When the female picks her roes and transfers them into her mouth, the male will shake his anal fins and show off the roe spots. The female, assuming they are her roes, proceeds to pick them off of the anal fins. At this moment, the male will release his milt directly into her mouth and, thus, effectively fertilize the roes. The female will then continue to hold the roes in her mouth. The mouthbrooding cichlids have a special throat pouch known as the buccal cavity, where the female stores her roes until they hatch. Between 21 and 36 days is the average length of time that the female cichlid holds the fry in its mouth, and it does not eat during this period. The female cichlid will continue to care for her fry for some time after hatching. For instance, she will hide it inside her mouth if there is danger nearby [14].

### 2.2. Experimental Design and Artificial Breeding

The experiment was performed after approval by the Ethics Commission of the University of Veterinary Medicine and Pharmacy in Košice (protocol code EKV/2022-11). Eight adult females and four males of the African cichlid species (*Labidochormis caeruleus*) (and two years of age) with sizes ranging from 11 to 13 cm were used in this study. The fish were kept in 240 L aquariums, and four females and two males were distributed in each tank, with a water temperature of 24 °C, hardness > 15 °dGH, and a pH of 8.0. They were fed twice a day with a complete mixture of Sera flora (Heinsberg, DE, Germany), and every fourth day, ¼ of the water in both aquariums was changed (Figure 1).

In order to introduce new methods of roe incubation and fry rearing, female cichlids that had roes stored in their buccal cavities were caught and removed from the tank, and a stripping method was used to release the roes early from the females. In this method, the buccal cavity was rinsed several times using a syringe filled with water. This was done over a bowl of aquarium water so that no roes remained in the cavity after the last rinse. Subsequently, the roes were transferred to an incubator with an artificial hatchery created by a pump and a cup with constant swirling of water and mixing of the roes through a filter tube.

The incubator served as a substitute for the mother’s mouth, so the roes were not in danger of being swallowed or spat out in a case of stress. From the washed-out roes, 4 groups were formed for the artificial incubation of the roes and rearing of the fry. The roes were selected at random and placed in groups of 40, with each group consisting of 2 replicates and 20 roes within each replicate. Subsequently, the roes were incubated in 1%, 5%, and 10% concentrations of humic acids, and the control group was not exposed to HA.

After 5–7 days of incubation, when the roes developed into fry, they were transferred from the incubator to aquariums with the same water volume of 10 L and HA concentrations, equipped with an air stone, and externally heated to the desired temperature of 26 °C by storing a water tank behind aquariums with fry and warming the water by 2 °C. After 30 days, all groups were humanely killed in water saturated with minerals and fixed in formalin, according to Wang et al. [15], for subsequent detection of physiological parameters (Figure 2).

### 2.3. Preparation of Alginite Concentrate with HA

Pre-dried, ground alginite (Algivo, s.r.o., Lučenec, Slovakia) of grain size 1–1.3 mm subjected to gamma-irradiation (Bioster, Veverská Bitýška, Czech Republic) was used as a fossil additive for the preparation of the concentrate. According to the method of Barančikova and Litavec [16], the alkaline solution (pH = 13) Na_4_P_2_O_7_ was used for the extraction of humic substances from alginite. This compound forms with humates of calcium and sesquioxides hydrated Ca and Fe diphosphates, insoluble in water but soluble in an excess of diphosphate, with the simultaneous formation of complex salts. Subsequently, the determination of the 72% share of humic acids from the prepared leachate was carried out at the Research Institute of Soil Science and Nature Conservation (Prešov, Slovakia) according to the Barančíková method [17].

### 2.4. Laboratory Analysis

The formalin-fixed cichlid fry from all four groups (12 fry from each group) had their body lengths measured at the Department of Morphological Disciplines, University of Veterinary Medicine and Pharmacy in Košice in histological sections. The body length from the gills to the beginning of the tail was measured on a microtome (Laika histoslide 2000) in the image J 1.45j program. The absorbance of the yolk sac in the monitored fry was evaluated during the last 8 days of the experiment, on the 22nd, 24th, 26th, 28th, and 30th days. Yolk sac absorbance was assessed visually, and it was considered absorbed if it was aligned with the ventral line of the gut and the fish started feeding actively.

The physical and chemical parameters of the water were checked every 5 days throughout the 30-day experimental period. The water temperature was monitored with a mercury aquarium thermometer (Sera, Heinsberg, DE). The levels of pH, general hardness (°dGH), nitrite, and nitrate were determined using Sera Quick tests (Heinsberg, DE) according to the method by Riffel et al. [3].

### 2.5. Statistical Analysis

The measured values of body length, yolk sac absorbance, and physicochemical parameters of the water were evaluated based on ANOVA variance analysis at the significance level α = 0.05. The control group was compared to groups with different concentrations of HA.

## 3. Results and Discussion

The results of the study showed a positive effect of the addition of HA on all three groups with respect to the increased viability of the roes and the further development of the fry during the 30-day period of monitoring in the artificial incubator. At the beginning of the study (on the first day), 40 roes were incubated in each group. On the fifth day (after fry hatching), the highest mortality rate was recorded in the control group (30%). On the same day, mortality in the group with the addition of 1% HA reached 17.5%, while in the groups with the addition of 5% and 10% HA, the recorded mortality rate was 5%. At the end of the monitored period (30 days), the highest mortality rate (40%) was recorded again in the control group. In the group with the addition of 1% HA, the recorded mortality was at the level of 20.0%; in the groups with the addition of 5% and 10% HA, the mortality levels were 7.5% and 10.0%, respectively. Based on ANOVA variance analysis at the α = 0.05 significance level, it was found that 5% and 10% concentrations of humic acids had a statistically significant effect on the survival of cichlids during the monitored period (Table 1).

The results of our study indicated a similar effect as that achieved by Meinelt et al. [18], who tested humic substances (HS) regarding acriflavine toxicity to embryos and larvae of the zebrafish Danio rerio. The authors modeled the aquatic environment for the fry with the addition of acriflavine. In the tested groups without the addition of HA, increased fry mortality was recorded compared to the groups that were given different concentrations of humic acids.

A significant effect of reduced mortality was noted by the author Abdel-Wahab et al. [19], who supplemented the feed of carp (*Cyprinus carpio*) with 0.4%, 0.8%, and 1% humic acids for 45 days. At the end of the monitoring period, the authors exposed all fish to nitric oxide at a dose of 1.75 mg/L for 5 days. In the groups supplemented with 0.8% and 1% humic acids, they recorded a mortality rate 50% to 70% lower than the mortality rate in the control group without the addition of humic acids.

Another indicator of the positive effect of the addition of HA was the mean length of the fry. The results indicate longer fry lengths in groups exposed to 5% and 10% concentrations of humic acids at the end of the 30-day monitoring period (Figure 3). The longer bodies of the fry in these groups were probably related to the acceleration of growth, with the resorption of the yolk sacs in all three groups exposed to 1%, 5%, and 10% HA concentrations being faster than that of the control group (Table 2). Based on statistical analysis, a higher number of fry with absorbed yolk sacs between 26 and 28 days was recorded in all groups exposed to HA compared to the control.

Table 3 shows the level of physical (temperature) and chemical (pH, nitrite and nitrate content, and water hardness) parameters in the water. There were obvious changes in the chemical properties of the water, especially between the control group and the groups with 5% and 10% concentrations of HA. Supplementation of HA affected water hardness, especially at the 5% and 10% concentrations, which is a positive feature when keeping fish species originating from habitats with soft water or during the artificial incubation and rearing of fish, such as in our case. Mungkung et al. [20] confirmed in their study that the addition of 0.5, 5.0, and 50.0 mg/L of humic acid resulted in a decrease in total hardness in soft, moderately hard, and hard-type water. In addition, humic substances had a favorable effect on the higher survivability of *Puntius gonionotus* after the addition of cadmium during the monitored period.

Another parameter that changed significantly was the pH of the water. The results showed a decrease in pH values, particularly within groups with 5% and 10% concentrations of HA, which was positively reflected in the lower mortality of moldy roes and fry due to the environment in which the molds cannot survive.

Perminova et al. [9] documented that HAs affect the functionality of the immune system and the growth of fish, detoxify heavy metals and organic pollutants, suppress the growth of cyanobacteria, regulate radiation function, and protect fish from the consequences of environmental stress. In water with a natural occurrence of humic acids, there are smaller fluctuations in oxygen concentrations and pH changes, which positively affect the adaptability of fish and reduce mortality during extreme weather changes. In their study, the authors also confirmed that HAs have anti-fungal, anti-parasitic, and antibacterial properties, inhibiting the growth of *Aeromonas hydrophila*, *A. sobria*, *Edwardsiella iclaluri*, *E. tarda*, *Pseudomonas fluorescens*, and *Escherichia coli*.

In natural conditions, pH values that are optimal for cichlids range from 7.7 to 8.6 for Lake Malawi and from 7.3 to 8.0 for Lake Tanganyika. Reported hardness values for Malawi range from 6 to 10 degrees of hardness (DH) and, for Lake Tanganyika, from 10 to 12 DH [14]. Experience has shown that both Tanganyikan and Malawian cichlid fish will prosper and breed at pH values as low as 7.2 and hardness values of 3 DH [21], which was also confirmed in our study. What seems critical is the stability of these values rather than their absolute magnitude. The results of our study show that 5% and 10% humic acid concentrations effectively stabilized the pH and hardness values during the monitored period.

One of the most important parameters in an aquarium that affects the viability of fish is the nitrite content of the water. A concentration of nitrite up to 0.1 mg/L is considered harmless to fry and young fish. At higher concentrations, there is a risk of acute poisoning and sudden death of the fry [2]. From the changes in the nitrite concentration in the monitored groups, it is clear that at 1% concentrations, HA was already able to bind the nitrites present in water and, thus, contributed to the successful rearing of fish and the stabilization of the environment.

The last of the important parameters, the concentration of the nitrates, is not as important as the concentration of the nitrites but still remains an important indicator in the evaluation of water quality [1]. The results of our study point to an additional positive effect of the supplementation of HA, namely its ability to reduce the concentration of nitrates in water. The optimal recommended nitrate concentration is 10–30 mg/L. Nitrates are not as harmful to fish as nitrites; therefore, even higher concentrations of 30–50 mg/L do not cause poisoning [2]. However, at very high concentrations (more than 50 mg/L), they cause acute poisoning, similar to nitrites. This was also confirmed in our study, as a high concentration of nitrates in the water (100 mg/L) in the control group most likely contributed to the 40% mortality rate in the fry.

## 4. Conclusions

The results of this study showed an increase in the successful artificial rearing of African cichlids (*Labidochormis caeruleus*) due to the creation of a suitable environment during the artificial incubation of roes and the development of fry after the addition of HA at different concentrations. The most significant changes in the stabilization of the aquatic environment occurred in groups exposed to 5% and 10% HA concentrations, namely with respect to the reduced levels of nitrites and nitrates, which significantly reduced the mortality of roes and increased the viability of fry. This knowledge can be used in the prevention of acute poisoning during the rearing of roe and fry in aquariums. In addition, the pH and hardness values of water decreased in a concentration-dependent manner as a result of the action of HA. Despite the fact that endemic cichlids from Lake Malawi prefer pH values from 7.7 to 8.6, fry can prosper and breed at pH values from 7.2 to 7.5 and hardness values from 7 to 10 DH. If the pH of the water reaches the alkaline range, the risk of roes or fry becoming moldy increases; therefore, with the influence of HA, we can reduce this risk and stabilize the pH values to a level where there are no losses. What seems critical is the stability of these values rather than their absolute magnitude. The results of our study show that 5% and 10% additions of humic acid concentrations effectively stabilized the pH and hardness values during the monitored period.

In addition to the stabilization of the physicochemical parameters of the water, differences in the size of the fry were noted in groups exposed to 5% and 10% HA concentrations. The body lengths of fry in these groups were larger at the end of the monitoring period compared to those of the control group, which confirmed the positive effect of HA on these organisms from a physiological point of view and in terms of the acceleration of growth through the absorption of the yolk sac. A significant difference was observed on the 26–28th days of the monitoring period in the fry in the groups with the addition of 5% and 10% of HA, when a 90% absorption rate of yolk sacs was observed compared to 50% in the control group.

The knowledge obtained in this study and its transfer into practice can help even less experienced aquarists successfully breed aquarium fish species that would not be able to survive under artificial breeding conditions without the addition of HA.

## Figures and Tables

**Figure 1 life-13-01071-f001:**
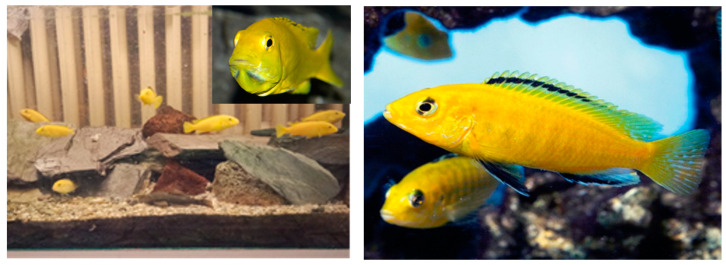
Aquarium with breeding cichlids, *Labidochormis caeruleus*, and a female throat pouch known as the buccal cavity, where the female stores her roes.

**Figure 2 life-13-01071-f002:**
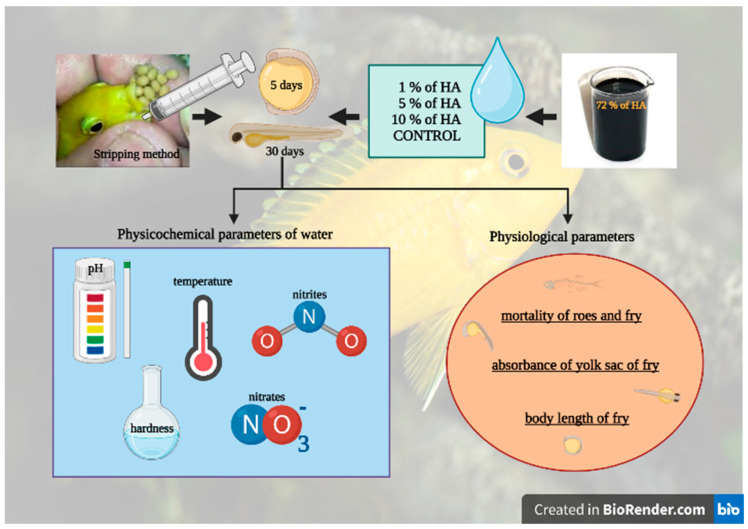
The schematic of the experimental design. It was created with BioRender.com.

**Figure 3 life-13-01071-f003:**
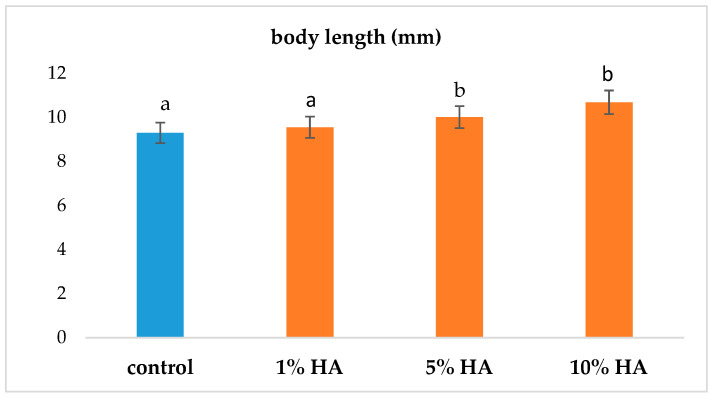
Comparison of body length of the fry at the end of the 30-day monitoring period. Note: a, b—values above the column with different letters differed significantly at *p* < 0.05.

**Table 1 life-13-01071-t001:** Mortality of roes and fry in the monitored groups during the 30-day monitoring period.

Day/Group	1% HA	5% HA	10% HA	Control
	Alive/Dead (n)	Alive/Dead (n)	Alive/Dead (n)	Alive/Dead (n)
Roe at 1st day	40/0	40/0	40/0	40/0
Roe at 5th day	33/7	38/2	38/2	28/12
Fry at 10th day	33/7	38/2	38/2	26/14
Fry at 20th day	33/7	38/2	36/4	26/14
Fry at 30th day	32/8	37/3	36/4	24/16
*p*-value	0.13	0.012 *	0.017 *	-

Note: The 1%, 5%, 10% HA—groups exposed to humic acids (HA) in various concentrations obtained by extraction from alginate; n—number of roe or fry in each monitored group. *p*-value *—based on ANOVA variance analysis at the significance level α = 0.05, the control group was compared to groups with different concentrations of HA.

**Table 2 life-13-01071-t002:** Comparison of yolk sac resorption during artificial incubation.

Day/Group	1% HA	5% HA	10% HA	Control
	n/%	n/%	n/%	n/%
Fry at 22th day	0/0	0/0	0/0	0/0
Fry at 24th day	0/0	6/15.8	11/30.5	0/0
Fry at 26th day	14/42.4	28/75.7	26/72.2	7/26.9
Fry at 28th day	23/71.8	34/91.9	33/91.7	12/50.0
Fry at 30th day	32/100.0	37/100.0	36/100.0	24/100.0
*p*-value	0.037 *	0.021 *	0.028 *	-

Note: The 1%, 5%, 10% HA—groups exposed to humic acids (HA) in various concentrations obtained by extraction from alginate; n—number of fry with resorbed yolk sac in each monitored group. *p*-value *—based on ANOVA variance analysis at the significance level α = 0.05, the control group was compared with groups with different concentrations of HA.

**Table 3 life-13-01071-t003:** Comparison of physicochemical properties of water.

Parameter/Group	1% HA	5% HA	10% HA	Control
Temperature (°C)	26 ± 1	26 ± 1	26 ± 1	26 ± 1
Hardness (°dGH)	15 ± 2.0 ^a^	10 ± 3.0 ^b^	7 ± 3.0 ^b^	16 ± 2.0 ^a^
pH	8.0 ± 0.5 ^a^	7.5 ± 0.5 ^b^	7.2 ± 0.6 ^b^	8.5 ± 0.5 ^a^
Nitrite (mg NO_2_/L)	1.0 ± 0.4 ^b^	0.1 ± 0.05 ^b^	0.1 ± 0.05 ^b^	5 ± 0.9 ^a^
Nitrate (mg NO_3_/L)	30 ± 5.5 ^b^	20 ± 2.0 ^b^	10 ± 1.5 ^b^	100 ± 20.0 ^a^

Note: The 1%, 5%, 10% HA–groups exposed to humic acids (HA) in various concentrations obtained by extraction from alginate. ^a,b^—values in the row with different letters differed significantly at *p* < 0.05.

## Data Availability

Not applicable.

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
