# Peer review of "The Effects of Humic Acids on the Early Developmental Stages of African Cichlids during Artificial Breeding"

_life, 2023, doi:10.3390/life13051071_

Round 1
Reviewer 1 Report
The manuscript titled 'The Influence of Microenvironment on Roes Incubation and Fry Development in Selected Species of Aquarium Fish', presents interesting results of a study indicating the effectiveness of egg disinfection methods used by aquarists. However, before publishing, the authors should make a number of corrections, which are listed below:
- please change the title - the paper is about the specific effects of humic acids on the early developmental stages of Labidochromis caeruleus
- lines 96-109 - is this description necessary for Materials and Methods? Rather, here it would be useful to describe the measurements of the fish taken for the experiment (age, body length, etc.).
- lines 156 - 163 - please describe what specific measurements were taken
Figure 2 - how was the 'absorbance of yolk sac' assessed? In Materials and Methods there is no description of how this measurement was made.
line 186 - in the cited article, the study looked at carp with a body length of about 10cm. Are there no other studies that indicate the usefulness of humic acids in aquaculture?
lines 194-202 - was it total length or body length?
Table 2 - how was the percentage of yolk sac resorption estimated?
line 162 and Table 3 - what hardness was the determination in the experiment - total/general (GH) hardness or carbonate hardness (KH)
lines 259-263 - was a safe and optimal pH level determined for endemic cichlids from Lake Malawi? In the wild, these fish live in alkaline waters.
Author Response
We thank the reviewer for their helpful comments and suggestions. We largely agree with the points raised and considered all of them in our revised manuscript. The point-to-point responses are listed in the following, and we have also highlighted the changes made in the text. The comments of the Editor and reviewers in the attachment are in italics and blue color, which are followed by our responses,
Yours sincerely.
(

Reviewer 2 Report
General appraisal:
The work was generally well conducted but the statistical treatment of the data should be improved (see specific comments) before publication.
Specific comments:
Statistical tretament of the data: Only the values of body length were statistically compared. However, all the other variables tested should be statistically compared also (biological and physico-chemical parameters).
Tables: Use always the same significant numbers for the same parameter.
Author Response
We thank the reviewer for their helpful comments and suggestions. We largely agree with the points raised and considered all of them in our revised manuscript. The point-to-point responses are listed in the following, and we have also highlighted the changes made in the text. The comments of the editor and reviewers in the attachment are in italics and blue color, which are followed by our responses,
Yours sincerely.
